# Assessing the Efficacy of Dietary Selenomethionine Supplementation in the Setting of Cardiac Ischemia/Reperfusion Injury

**DOI:** 10.3390/antiox8110546

**Published:** 2019-11-13

**Authors:** Leila Reyes, David P. Bishop, Clare L. Hawkins, Benjamin S. Rayner

**Affiliations:** 1Heart Research Institute, Sydney 2042, Australia; leila.reyes@hri.org.au (L.R.); clare.hawkins@sund.ku.dk (C.L.H.); 2Sydney Medical School, University of Sydney, Sydney 2006, Australia; 3School of Mathematical and Physical Sciences, Faculty of Science, University of Technology Sydney, Sydney 2007, Australia; David.Bishop@uts.edu.au; 4Department of Biomedical Sciences, Panum Institute, University of Copenhagen, DK-2100 Copenhagen, Denmark

**Keywords:** ischemia/reperfusion (I/R), hypochlorous acid (HOCl), myeloperoxidase (MPO), selenomethionine (SeMet)

## Abstract

Oxidative stress is a major hallmark of cardiac ischemia/reperfusion (I/R) injury. This partly arises from the presence of activated phagocytes releasing myeloperoxidase (MPO) and its production of hypochlorous acid (HOCl). The dietary supplement selenomethionine (SeMet) has been shown to bolster endogenous antioxidant processes as well as readily react with MPO-derived oxidants. The aim of this study was to assess whether supplementation with SeMet could modulate the extent of cellular damage observed in an in vitro cardiac myocyte model exposed to (patho)-physiological levels of HOCl and an in vivo rat model of cardiac I/R injury. Exposure of the H9c2 cardiac myoblast cell line to HOCl resulted in a dose-dependent increase in necrotic cell death, which could be prevented by SeMet supplementation and was attributed to SeMet preventing the HOCl-induced loss of mitochondrial inner trans-membrane potential, and the associated cytosolic calcium accumulation. This protection was credited primarily to the direct oxidant scavenging ability of SeMet, with a minor contribution arising from the ability of SeMet to bolster cardiac myoblast glutathione peroxidase (GPx) activity. In vivo, a significant increase in selenium levels in the plasma and heart tissue were seen in male Wistar rats fed a diet supplemented with 2 mg kg^−1^ SeMet compared to controls. However, SeMet-supplementation demonstrated only limited improvement in heart function and did not result in better heart remodelling following I/R injury. These data indicate that SeMet supplementation is of potential benefit within pathological settings where excessive HOCl is known to be generated but has limited efficacy as a therapeutic agent for the treatment of heart attack.

## 1. Introduction

Strong evidence has shown a role of infiltrating leukocytes in mediating host tissue damage during and following cardiac ischemia/reperfusion (I/R) injury [1,2,3,4], which may in part be due to their role in contributing to oxidative stress within this environment [5]. Activated leukocytes, in particular neutrophils, are both a source of the oxidants superoxide (O_2_•^−^) and hydrogen peroxide (H_2_O_2_), as well as releasing myeloperoxidase (MPO), which drives the formation of hypochlorous acid (HOCl). All of these oxidants share the same trait of being capable of causing the cardiac myocyte cellular dysfunction characteristic of cardiac I/R injury, including impaired mitochondrial function and disruption to calcium (Ca^2+^) regulation, resulting in cell death and contributing to the phenomenon of myocardial stunning [6,7,8,9].

Given that oxidative stress has been established as a prominent feature of cardiovascular disease (CVD) and owing to their demonstrated ability to act as potent scavengers of radical species [10,11,12,13], antioxidant administration remains a potentially beneficial therapy within this setting. Despite this knowledge and the fact that prior benefits have been demonstrated within in vitro settings, particularly through the use of a variety of vitamins and polyphenolic compounds [11], the vast majority of human clinical trials involving antioxidant supplementation have demonstrated a lack of benefit of these agents in the prevention of CVD [14]. This mainly stems from the challenge of achieving a high enough concentration of the antioxidant therapeutic in the correct location or cellular compartment to compete with the more abundant cellular targets.

Selenium is an essential trace element that supports numerous important cellular functions ranging from the synthesis of seleno-dependent antioxidants, including glutathione peroxidase (GPx) and thioredoxin reductase (Trxrd), responsible for the maintenance of intracellular redox status, through to processes involved in selenium transport [15]. Selenium species offer a promising, alternative therapy against oxidative stress within the setting of CVD due to their ability to rapidly react with a variety of biologically-relevant oxidants [16,17,18], including H_2_O_2_ and HOCl, at rates comparable to the reaction of these oxidants with primary cellular targets such as abundant low molecular-weight and protein thiols (R-SH) species [19], whilst also promoting the activity of endogenous seleno-dependent antioxidants, particularly those of the GPx and Trxrd systems [20,21,22,23,24,25,26]. In addition, recent studies highlight a potential catalytic role for selenium-containing compounds, including selenomethionine (SeMet) to scavenge oxidants [27,28].

The benefits of selenium supplementation in improving the body’s selenium status has been highlighted in numerous selenium-deficient animal models as well as human trials [29,30,31,32]. In particular, therapeutic selenium administration has been shown as beneficial in several in vivo disease models where oxidative stress is prominent, including atherosclerosis [33,34,35,36] and various I/R pathologies such as renal and liver I/R injury [37,38]. The promise of selenium supplementation as a suitable therapeutic agent for the prevention of CVD is shown by the association of low selenium levels with the incidence of CVD [39,40,41] with low plasma concentrations of selenium associated with poor clinical outcome, accompanying future CVD deaths in ACS patients [40]. In vitro, the effect of a variety of selenium species, most notably, SeMet and sodium selenite has been previously investigated in a variety of cell types including cardiac cells [20,42,43,44]. In these studies selenium supplementation was shown to protect against cellular damage arising from different models of oxidative stress, including exposure to physiologically relevant concentrations of the oxidant H_2_O_2_ [21,26]. Despite this wealth of data there is scant information regarding the ability of selenium species to protect against the oxidation-induced cellular damage that is observed specifically within the setting of cardiac I/R injury. Much of the current data relating to the effects of selenium supplementation in cardiac injury models have been limited to selenium-deficient animals and/or ex vivo models of cardiac injury [45,46,47], with limited data relating to the efficacy of selenium supplementation in preventing the longer-term adverse remodelling of heart tissue and consequent cardiac dysfunction evident in late stage cardiac I/R injury or heart failure (HF).

The organic seleno-compound SeMet is a highly favourable candidate amongst the range of selenium species of potential benefit for antioxidant-based therapeutics due to it being more readily taken up by cells and incorporated into cellular proteins in place of the essential amino acid methionine [22,23,48,49]. Once present and incorporated, SeMet is able to react with oxidants and then be catalytically recycled back to an active form by low molecular environmental reductants and intracellular reducing systems [50,51] including glutathione reductase (GR) and Trxrd. In this respect, the antioxidant capacity of SeMet is continually regenerated at the cellular level [27,28,51]. Given this, the aims of the current study were to evaluate the potentially protective role of SeMet supplementation in modulating the extent of cellular damage observed in an in vitro cardiomyocyte model exposed to HOCl and an experimental in vivo model of cardiac I/R injury.

## 2. Materials and Methods

### 2.1. Chemicals and Reagents and Quantification of Hypohalous Acids

All aqueous solutions and buffers were prepared with nano-pure water (npH_2_O) filtered through a four-stage Milli Q system (Millipore-water, Lane Cove, NSW, Australia). All chemicals unless otherwise stated were purchased from Sigma-Aldrich (St. Louis, MO, USA). Working concentrations of HOCl was prepared from a concentrated stock solution of NaOCl (Labchem Pty Ltd., Johannesburg, Gauteng, South Africa).

### 2.2. Cell Culture

All experiments were performed with the rat cardiac myoblast H9c2 cell line (ECACC, Salisbury, UK). Cells were maintained in Dulbecco’s modified Eagle’s medium (DMEM) supplemented with 10% (v/v) foetal bovine serum (FBS, Bovogen Biologicals Pty Ltd., East Keilor, VIC, Australia) and 2 mM l-glutamine (Lonza, Basel, Switzerland) in 175 cm^2^ tissue culture flasks at 37 °C in a humidified atmosphere containing 5% CO_2_. For experiments, cells were seeded at a density of 1 × 10^5^ cells mL^−1^ in 12-well plates at a volume of 1 mL unless otherwise stated and left to adhere overnight.

### 2.3. In Vitro SeMet Supplementation and Oxidative Insult

SeMet supplementation in vitro was achieved by exposing cells to SeMet (25 µM) prior to and during oxidative insult, i.e., exposure to HOCl. Following seeding, cells were washed with warmed (37 °C) Hank’s buffered salt solution (HBSS) and serum-starved in the presence or absence of SeMet (25 µM) for 24 h. Cells were then exposed to HOCl by first washing the cells with HBSS to ensure any reactions of either HOCl with media components would not confound results [52,53], and then exposed to HOCl (40–50 µM) in the presence or absence of SeMet (25 µM) in HBSS. Subsequent analysis of the cells was performed immediately following oxidant exposure.

### 2.4. Animals and Diets

This study was approved by the local ethics committee (Sydney Local Health District (SLHD) Animal Welfare Committee; Protocol Number: 2015-008A). Male Wistar rats (100–125 g) were supplied by the Animal Resource Centre (Perth, WA, Australia). Animals were housed at 20–24 °C and under a 12 h light/dark cycle with free access to food and water and were given at least one week for acclimatisation. Rats were then randomly assigned into groups receiving either normal chow (control; Specialty Feeds, Glen Forrest, WA, Australia) or normal chow supplemented with 2 mg kg^−1^ SeMet ad libitum for 8 weeks. The dose of SeMet administered equated to ~50 µg SeMet per day as assessed over the course of the study (Appendix A), chosen to mimic human dietary intervention [54]. Animals assigned normal chow were given the same amount of feed as the SeMet-supplemented group. Rats were weighed, and their health and food intake were recorded on a weekly basis, and their general wellbeing and availability of food was visually monitored daily.

### 2.5. In Vivo Cardiac Ischemia/Reperfusion

Following the 8 week feeding regime, animals were placed under general anaesthetic by placing the animal in an induction chamber ventilated with 5% (v/v) vaporised isoflurane. The animals were intubated and the ventilation (tidal volume: 1.5 mL/100 g of body weight; ventilation rate, 75 strokes min^−1^) and anaesthesia maintained with 2% (v/v) isoflurane for the duration of the surgery. Immediately prior to surgery, an intramuscular injection of lignocaine (10 mg kg^−1^ body weight) was administered. A left thoracotomy was performed, and the pericardium opened to expose the heart. A topical application of lidocaine was then applied directly to the heart to alleviate the potential for arrhythmia. Ischemia was induced by creating a transition ligation using a snare. This involved passing a 6–0 silk suture under the left anterior descending (LAD) artery, threading the suture through a small 1 mm piece of polyethylene tubing and clamping the tubing using a hemostat. Following 30 min, the snare was released, and the stitch removed to allow for reperfusion. The chest cavity was closed, and subcutaneous lidocaine was applied to the wound site, with the animal revived and placed in an isolated chamber for recover for 1–2 h and returned to their original cage. At this point, the animal was given jelly containing 10% buprenorphine solution for long-term pain relief. For the sham cohort, the operative technique was identical except that no ligation of the LAD was performed.

### 2.6. Flow Cytometric Analysis

Analysis of cell death, mitochondrial inner transmembrane potential (ΔΨ_m_) and cytosolic Ca^2+^ concentrations in cells exposed to HOCl and supplemented with or without SeMet were performed immediately post treatment using flow cytometry, with flow analysis of the samples conducted using a BD FACSVerse™ flow cytometer (BD, North Ryde, NSW, Australia). For the analysis of cell death, Annexin-V APC (BD, North Ryde, NSW, Australia) and propidium iodide (PI) was used to detect apoptotic and necrotic cell populations, respectively. Briefly, cells were washed with phosphate buffered saline (PBS) and harvested with trypsin/EDTA. Cells were then resuspended in binding buffer (10 mM HEPES/ NaOH (pH 7.4), 140 mM NaCl and 2.5 mM CaCl_2_), incubated with Annexin-V APC and PI in the dark at 21 °C for 20 min and run on the flow cytometer. ΔΨ_m_ was assessed using the MitoProbe JC-1 assay kit as per the manufacturer’s instructions (Molecular Probes, Eugene, OR, USA), with stock solutions (200 µM) prepared using dimethyl sulfoxide (DMSO) and final concentrations (2 µM) prepared in HBSS. For this assay, cells were washed with PBS and harvested with trypsin/EDTA, incubated with JC-1 in HBSS in the dark at 21 °C for 30 min and run on the flow cytometer. Cytosolic Ca^2+^ concentrations were measured using the Fluo,4-AM Ca^2+^ probe (Life Technologies, Carlsbad, CA, USA), with stock solutions (1 mM) prepared in DMSO and the final concentration (5 µM) prepared in Ca^2+^ supplemented HBSS. Cells were prepared for analysis by first washing the cells with HBSS followed by incubation with Fluo,4-AM in HBSS at 37 °C/5% CO_2_ for 45 min. Cells were subsequently detached from tissue culture wells by gently scraping the cells with a cell scraper and run on the flow cytometer.

### 2.7. Glutathione Peroxidase Activity Assay

GPx activity was indirectly measured in cells exposed to HOCl and supplemented with or without SeMet and the plasma of the control and SeMet-supplemented animals by measuring NADPH consumption in the presence of GR and glutathione (GSH) during the reduction of *tert*-butyl hydroperoxide (*tert*-Bu-OOH), as first described by Flohé et al. [55]. In brief, cell lysates of plasma samples were incubated in a GPx reaction buffer (50 mM sodium phosphate buffer, pH 7.4, 5 mM EDTA, 8.8 mM GSH, 0.5 U baker’s yeast GR, 0.5 mM NADPH) and 0.1 mM *tert*-Bu-OOH. NADPH consumption was monitored spectrophotometrically at 340 nm for 15 min at 1 min intervals at 37 °C, with GPx activity determined by a change in absorbance at 340 nm over time and expressed as the percentage of the control.

### 2.8. Cell Proliferation Assay

Cell death in cells exposed to HOCl and different forms of SeMet supplementation was performed immediately following oxidant exposure using the CellTiter 96^®^ AQueous One Solution Cell Proliferation Assay from Promega (Madison, WI, USA). For this assay, H9c2 cells were seeded at a density of 1 × 10^5^ cells mL^−1^ in 96-well plates at a volume of 100 µL and left to adhere overnight. Cells were then either exposed to SeMet in serum-free media for 24 h prior to and during exposure to HOCl for 1 h (SeMet group), serum-starved for 24 h in the presence of SeMet and then exposed to HOCl alone for 1 h (pre-treatment group), or serum-starved for 24 h and exposed to HOCl and SeMet simultaneously for 1 h (co-treatment group) at the doses indicated in the figure legends. Following oxidant exposure, cells were washed with warmed PBS and incubated in complete DMEM (100 µL/well) containing the CellTiter 96^®^ AQueous One Solution Reagent (10 µL/well) at 37 °C/5% CO_2_ for 4 h. The absorbance was measured at 490 nm using a plate reader, with the final values expressed as a percentage of the control.

### 2.9. Lactate Dehydrogenase (LDH) Release Assay

Using the same experimental set-up for cell culture as described in the previous section, further confirmation of cell death was analysed 24 h following oxidant exposure using the LDH release assay. Supernatant was collected and centrifuged at 2000× *g* for 5 min to remove any cellular debris that would interfere with absorbance. Cell lysates (10 µL) and supernatants (10 µL), and blanks containing media or npH_2_O alone, were incubated with a reaction buffer containing 0.15 mg mL^−1^ NADH and 2.5 mM sodium pyruvate (200 µL), with NADPH consumption monitored spectrophotometrically at 340 nm for 30 min at 5 min intervals using a plate reader. The cellular viability was calculated using the following equation after first removing the background LDH activity:
$$\mathrm{Viability}\;(\%)\;=\;\frac{\Delta \;\mathrm{intracellular}\;\mathrm{LDH}\;\mathrm{activity}}{\Delta \;\mathrm{intracellular}\;\mathrm{LDH}\;\mathrm{activity}\;+\;\Delta \;\mathrm{extracellular}\;\mathrm{LDH}\;\mathrm{activity}}\;\times \;100.$$

### 2.10. Quantification of Cellular Thiols

The fluorometric method utilising ThioGlo 1 (Berry & Associates Inc, Dexter, MI, USA) was used to quantify cellular thiols in HOCl-exposed and SeMet-supplemented cells as described by Hawkins et al. [56]. Briefly, following oxidative insult and SeMet supplementation, cells were washed with warmed HBSS and lysed in ice-cold npH_2_O. Samples were then incubated with the ThioGlo 1 reagent (13 µM) in the dark for 5 min at 21 °C and fluorescence was recorded at λ_ex_ 360 nm and λ_em_ 530 nm. Quantification of thiols was calculated using a standard curve constructed with GSH.

### 2.11. Quantification of GSH

GSH was quantified in cells exposed to HOCl insult with or without SeMet supplementation using monobromobimane with HPLC separation as detailed previously [57]. Briefly, cells were washed with warmed HBSS and lysed in 75 µL ice-cold npH_2_O and 75 µL KPBS buffer (50 mM potassium phosphate buffer, 17.5 mM EDTA, 50 mM serine, 50 mM boric acid, pH 7.4). γ-glutamylglutamine (0.1 mM in PBS) was added to all samples as an internal standard, and samples were then incubated with 10 µL monobromobimane (3 mM in acetonitrile) in the dark for 30 min. Perchloric acid (10 µL, 70% (v/v)) was added to stop the reaction and samples were filtered through 0.2 µm centrifugal filters (Millipore, Burlington, MA, USA). GSH was quantified after separation using a Shimadzu HPLC system equipped with a Synergi 4 µm Hydro-RP C-18 column (150 × 4.6 mm; Phenomenex, Lane Cove, NSW, Australia), maintained at 30 °C, with a flow rate of 1 mL min^−1^. Mobile phase A contained 1% (v/v) acetic acid and 5% (v/v) acetonitrile and mobile phase B consisted of 1% (v/v) acetic acid and 20% (v/v) acetonitrile, with the pH of both mobile phases adjusted to pH 4.5 using ammonium hydroxide. The derivatives of GSH and γ-glutamylglutamine were observed by fluorescence detection (RF10A-XL; Shimadzu, Rydalmere, NSW, Australia) at λ_ex_ 328 nm and λ_em_ 542 nm.

### 2.12. Echocardiography

A transthoracic echocardiogram was performed at 4 weeks post experimental/sham I/R injury prior to animals being euthanised and subsequent tissue harvesting. Animals were anaesthetised and positioned as described above. The exam was performed using the General Electric (GE) Vivid Q ultrasound machine (Milwaukee, WI, USA), using a 4–15 MHz broadband transducer (L8-18i-D). Recordings were viewed in M-mode to measure the left ventricular end-systolic diameter (LVESD) and left ventricular end-diastolic diameter (LVEDD), with measurements taken for three recordings and used to calculate fractional shortening (FS) and ejection fraction (EF).

### 2.13. Tissue Collection

Following the 8 week feeding regime, or 24 h or 4 weeks post experimental sham or I/R injury, animals were euthanised by first anaesthetising the animal as described above. The chest cavity was opened, and blood was extracted from the aorta before excising the heart. Blood was collected in blood collection tubes spray-coated with EDTA and was centrifuged at 200× *g* for 10 min at 21 °C for the collection of plasma. Once excised, the heart was flushed with saline and sliced in 2 mm sections. For uniform comparison between hearts, the first 2 mm section commencing from the heart apex was designated for RNA and protein analysis. The following 2 mm sections were designated for histological analysis.

### 2.14. Quantification of Selenium Using Inductively Coupled Plasma Mass Spectrometry (ICP-MS)

An Agilent Technologies 7500cx ICP-MS (Agilent Technologies, Mulgrave, Australia) was used with sample introduction via a micromist concentric nebuliser (Glass Expansion, West Melbourne, Australia) and a Scott type double pass spray chamber cooled to 2 °C. The sample solution and the spray chamber waste were carried with the aid of a peristaltic pump. ICP-MS extraction lens conditions were selected to maximise the sensitivity of a 1% HNO_3_:HCl solution containing 1 ng mL^−1^ of Li, Co, Y, Ce and Tl. Helium was added into the octopole reaction cell to reduce interferences. Calibration curves were constructed, and the results analysed using Agilent Technologies Masshunter software.

A certified selenium calibration standard, Seastar Baseline nitric acid (HNO_3_), and Seastar Baseline hydrogen peroxide (H_2_O_2_) were obtained from Choice Analytical, Thornleigh, Australia. Calibration curves were constructed from seven standards at 0, 1, 10, 50, 100, 500 and 1000 ng mL^−1^ in a matrix matched diluent.

Tissue (0.1 g) was collected from animals fed either a normal diet or a diet supplemented with SeMet (2 mg kg^−1^) for 8 weeks and was freeze-dried, weighed and then digested on a heat block at 70 °C in 0.25 mL HNO_3_ and 0.25 mL H_2_O_2_ in a pre-weighed microcentrifuge tube. After digestion, the solution was diluted with ultrapure water to an approximate volume of 2 mL and re-weighed to get the final digestion volume. Plasma samples were diluted 1:10 before analysis.

### 2.15. Triphenyl Tetrazolium Chloride (TTC) Staining

In a separate cohort of animals, 2,3,5-triphenyltetrazolium chloride (TTC) staining was used to differentiate viable and infarcted myocardium of animals subjected to sham injury or 30 min ischemia and 24 h reperfusion. Immediately after being flushed with saline, the heart was sliced in 2 mm sections and incubated in 1% (w/v) TTC in PBS for 20 min at 37 °C. The sections were transferred to 10% (v/v) formalin for better visualisation and imaged using a Nikon SMZ800 Zooming Body Microscope, with images stored as tiff files. The infarct size was measured by computerised planimetry using Image J and calculated as a percentage of the left ventricle for each 2 mm section.

### 2.16. Histological Studies

The second and third 2 mm sections commencing from the heart apex were fixed in 10% (v/v) formalin solution overnight and paraffin embedded. Then, 4 µm sections were de-paraffinized and rehydrated and stained using Milligan’s trichrome for the detection of fibrosis in the heart tissue of animals subjected to I/R injury, i.e., 30 min ischemia followed by reperfusion and recovery for 4 weeks or sham injury. Afterwards, slides were dehydrated, cleared and mounted, with imaging performed on a slide scanner Axio Scan.Z1. The extent of fibrosis was assessed by measuring the degree of blue staining from the Milligan’s trichrome stain using Image J. Images were first converted to a red-green-blue (RGB) image stack, with the red stack used to visually adjust the threshold setting so that only the stained area was highlighted. Threshold settings were adjusted only once for the first image and were applied for subsequent images. The selected stack was then measured using ‘Particle Analysis’, where the percentage area calculated was taken as the percentage area of the left ventricle (LV) stained with Milligan’s trichrome stain.

### 2.17. qPCR

RNA from H9c2 cells was extracted using the ReliaPrep™ RNA Cell Miniprep System from Promega (Madison, WI, USA) as per the manufacturer’s instructions. Reverse transcription was performed with the iScript cDNA Synthesis Kit (Bio-rad, Sydney, Australia) following the manufacturer’s protocol. mRNA gene expression was assessed by qPCR using the primer sequences outlined in Table 1 and iQ™ SYBR^®^ Green Supermix (Bio-Rad) with the following PCR conditions: 95 °C for 30 s, 60 °C for 30 s, 72 °C for 30 s for 39 cycles, followed by melt curve analysis. Data were normalised to the reference genes *Nono* and *β-actin* and presented as a fold-change compared to HBSS-treated control cells.

### 2.18. Statistical Analysis

All statistical analysis was performed using GraphPad Prism 7 (GraphPad Software, San Diego, CA, U.S.A., http://www.graphpad.com), with *p* < 0.05 taken as significant. Details of the tests performed for each experiment are given in relevant figure legends.

## 3. Results

### 3.1. SeMet Protects Against Cellular Damage Elicited by HOCl in H9c2 Cells

We have previously demonstrated that the concentration of HOCl required to elicit detrimental changes to H9c2 cells was between 40 µM and 50 µM [58]. The ability of SeMet to protect against cellular damage induced by these concentrations of HOCl in H9c2 cells was first assessed by examining mitochondrial function with flow cytometry and JC-1 staining. Exposure of H9c2 cells to HOCl for 1 h resulted in the loss of mitochondrial inner trans-membrane potential (ΔΨ_m_), which was negated when cells were supplemented with SeMet (25 µM) for 24 h prior to and during oxidative insult with 40 µM, but not 50 µM HOCl (Figure 1A,B). Under the same experimental conditions, cytosolic Ca^2+^ concentrations were measured using the fluorescent Ca^2+^ indicator, Fluo,4-AM for flow cytometry and demonstrated an increase in cytosolic Ca^2+^ in H9c2 cells exposed to HOCl. This result was not observed in cells supplemented with SeMet (25 µM), with intracellular Ca^2+^ levels remaining comparable with those shown in the HBSS control across all HOCl concentrations examined (Figure 1C,D). Given that SeMet protected against HOCl-induced cellular dysfunction, subsequent studies were performed to assess the extent of cell death with flow cytometry and Annexin/PI staining. Exposure of H9c2 cells to 40 µM HOCl for 1 h resulted in a significant increase in PI staining (*p* < 0.01), indicative of necrotic cell death, which was prevented with SeMet (25 µM), as demonstrated by a statistically significant decrease in PI staining compared to cells exposed to HOCl alone (Figure 1F). Similarly, H9c2 cells that underwent late apoptosis upon exposure to 50 µM HOCl as indicated by an increase in both Annexin-V and PI staining was ameliorated with SeMet (25 µM) (Figure 1G). However, SeMet was not able to prevent necrotic cell death in H9c2 cells exposed to 50 µM HOCl (Figure 1E,F). Given that selenium-containing compounds can exert cellular toxicity [21,22,23], we confirmed that no cell death was observed on supplementation with SeMet (0–25 µM) in the absence of oxidant treatment (Figure 2A).

### 3.2. Redox Status of H9c2 Cells Exposed to HOCl is Altered with SeMet Leading to Protection

Endogenous cellular antioxidant systems, in particular those centred around GPx, are essential for combatting oxidative stress, with the activity of GPx within cells dependent on the adequate levels of available selenium [59]. Preliminary studies assessing the GPx activity of H9c2 cells exposed to increasing concentrations of SeMet showed a dose-dependent increase in GPx activity, with statistical significance (*p* < 0.05) demonstrated upon exposure to ≥10 µM SeMet at 24 h compared to control cells assayed in the absence of SeMet (Figure 2B). The effect of SeMet supplementation on the regulation of seleno-dependent antioxidant enzyme mRNA expression was also assessed. Measurement of H9c2 mRNA expression of *GPx1* at 24 h following exposure to 25 µM SeMet demonstrated a slight, but not statistically significant, increase in gene expression (Figure 2C). However, there were no significant changes in thioredoxin (*Trx*)1 or *Trxrd1* mRNA expression in H9c2 cells incubated either in the presence or absence of SeMet supplementation (Figure 2C).

The ability of SeMet to alter GPx activity in H9c2 cells exposed to oxidative insult by HOCl was next examined as an indicator of the redox status of cells. H9c2 cells exposed to HOCl concentrations (≥40 µM) and supplemented with SeMet (25 µM) prior to and during oxidative insult demonstrated elevated GPx activity compared to cells exposed to HOCl in the absence of SeMet supplementation (Figure 2D), thus suggesting that the promotion of GPx activity by SeMet is the mechanism underlying the protection against HOCl-induced cellular damage.

Considering that SeMet was shown to elicit protection against HOCl-induced cellular damage, the mechanism of protection was next assessed in subsequent viability assay studies. Experiments were performed in cells exposed to HOCl alone (control), cells supplemented with SeMet throughout the whole assay period (SeMet), prior to the exposure of either 40 or 50 µM HOCl (pre-treatment) or only during HOCl exposure (co-treatment). Assessment of viability using MTS as a measure of metabolic activity demonstrated that all experimental conditions employing SeMet supplementation protected against H9c2 cell death in response to exposure to 40 µM HOCl, whilst only the continual supplementation with SeMet over the assay period (SeMet group) or supplementation with SeMet only during HOCl exposure (co-treatment group) afforded significant rescue of cell viability when measured immediately following exposure to 50 µM HOCl (Figure 2E). To further confirm this result, cellular viability at 24 h post HOCl exposure was assessed using the LDH release assay and demonstrated a protection against cell death in H9c2 cells exposed to 40 µM HOCl when cells were supplemented with SeMet throughout the assay period (SeMet) as well as only during exposure to HOCl (co-treatment group) (Figure 2F). H9c2 cells supplemented with SeMet only prior to the exposure of 50 µM HOCl (pre-treatment group) appeared to partially attenuate cell death, yet this result did not reach statistical significance (Figure 2F).

Using the experimental set-up as described in the section above, GPx activity was next assessed to further confirm the mechanism of protection offered by SeMet against the cellular damage elicited by HOCl in H9c2 cells. As shown in Figure 2G, supplementation only during the exposure to HOCl (co-treatment) protected against the loss of GPx activity in H9c2 cells exposed to ≥40 µM HOCl. In contrast, the supplementation of SeMet prior to oxidant exposure (pre-treatment) failed to rescue the loss of GPx activity across all HOCl concentrations examined (Figure 2F).

### 3.3. Profile of Selenium Content in Tissues of SeMet-Supplemented Rats

Given that the in vitro data demonstrated the ability of SeMet to protect against HOCl-induced cellular dysfunction characteristic of cardiac I/R injury, and the strong association of inflammation and neutrophil infiltration following myocardial infarction (MI), SeMet supplementation was next assessed in vivo. Male Wistar rats were randomly assigned normal chow (control group) or normal chow supplemented with SeMet (2 mg kg^−1^) (SeMet group) ad libitum for an 8 week period. Over the course of the feeding regime, there were no differences in the food consumption or the increases in body weight between the control and SeMet group (Appendix A).

The efficacy of SeMet supplementation in vivo was first assessed by examining the profile of tissue selenium. Quantification of selenium in the tissues harvested from animals of either the control or SeMet group was measured using inductively coupled plasma-mass spectrometry (ICP-MS) following 8 weeks of feeding. The level of plasma selenium was significantly higher in the SeMet group than in the control group, rising from basal levels in the control group of ~380 ng mL^−1^ to ~470 ng mL^−1^ in rats supplemented with SeMet (Figure 3A). In the heart, there was a ~2-fold increase in selenium within the tissue of the SeMet supplemented group compared to controls, from ~2000 ng g^−1^ to ~4000 ng g^−1^ (Figure 3B). Similarly, the ability of SeMet to cross the blood–brain barrier was demonstrated with a ~3-fold increase in brain selenium content in SeMet supplemented rats compared to rats in the control group (Figure 3C). There was no change in selenium content in the liver (Figure 2D) and a small, but significant increase in kidney selenium content within SeMet supplemented rats compared to controls (Figure 3E). Despite the increase in selenium evident within the plasma of rats on a SeMet supplemented diet, this did not equate to an increase in total GPx activity (Figure 3F).

### 3.4. Effect of SeMet Supplementation Following I/R Injury

In order to first examine the ability of SeMet to protect against cardiac I/R injury, we next assessed the degree of infarct was next examined in 2 mm transverse sections of heart tissue stained with triphenyl tetrazolium chloride (TTC) following 24 h reperfusion. I/R injury in both the control and SeMet-supplemented group produced infarcts that spanned the LV (Figure 4A), whilst there was no evidence of infarcts present in the LV of sham animals in either feeding groups (data not shown). Quantification of infarct size demonstrated a smaller infarct size in two of the heart sections in SeMet-supplemented animals subjected to I/R injury compared to controls, yet statistical significance (*p* < 0.01) was only achieved for the heart section midway from the apex, i.e., 6 mm (Figure 4B).

The pathology of late-stage I/R injury is characterised by impaired cardiac function, often arising from the adverse remodelling of the heart that involves responses that are characteristic of wound healing and scar formation [60]. To determine the effect of SeMet supplementation in modulating these responses, a separate cohort of rats recovered for 4 weeks following I/R injury. Cardiac function was assessed by echocardiography with representative echocardiogram images shown in Figure 4C. Diastolic (D) and systolic (S) measurements were recorded over time with rats subject to I/R injury displaying impaired cardiac function, as demonstrated by a statistically significant decrease in left ventricular fractional shortening (FS) of approximately 20% (Figure 4D) and corresponding ejection fraction (EF) of approximately 10% (Figure 4E) when compared to the sham cohort. Rats supplemented with SeMet demonstrated similar levels of cardiac dysfunction following I/R injury despite a slight, but not statistically significant, improvement in the EF and FS across all SeMet-supplemented animals compared to controls.

Chronic cardiac fibrosis is a prominent feature associated with adverse cardiac remodelling and involves the excessive deposition of extracellular matrix (ECM), primarily collagen, in the cardiac muscle [61]. Hence, the extent of fibrosis was determined by utilising Milligan’s trichrome staining to differentiate between collagen (blue staining) and muscle tissue (purple staining) in the infarcted myocardium (Figure 4F). Extensive cardiac fibrosis was observed in all rats subjected to I/R injury compared to sham-operated rats, with no difference in the extent of fibrosis evident between control and SeMet fed cohorts (Figure 4G).

## 4. Discussion

This study is the first to investigate the effect of SeMet against the oxidative cellular damage mediated by HOCl in vitro in cardiomyocytes, as well as its potential beneficial effect in an experimental in vivo rat model of cardiac I/R injury. Both the in vitro and in vivo data suggest that the protection elicited by SeMet is limited, with the in vitro data demonstrating the failure of SeMet to protect against the cellular dysfunction in cardiac myocytes exposed to 50 µM HOCl, yet SeMet was able to afford a ~50% protection against 40 µM HOCl-induced cardiac myocyte necrosis, concomitant with the reversal of ΔΨ_m_ and an increase in cytosolic Ca^2+^.

The current results are in strong contrast with studies that support the overall benefit in using selenium supplementation in the form of SeMet, as well as other selenium species, including sodium selenite and selenium nanoparticles, as a form of antioxidant therapy to protect against the cellular damage arising from different models of oxidative stress in various mammalian cell types [21,25,26]. This included rescuing cells from cell death [42,43], mitochondrial dysfunction [25,26,62] and Ca^2+^ overload [21], with further beneficial effects demonstrated by the reduction in markers of oxidative stress, such as improving antioxidant capacity [21] and decreasing reactive oxygen species (ROS) production [42,43] and lipid peroxidation [42]. The mechanism of protection in these studies was described to arise from either the promotion of antioxidant activity, in particular, GPx, induced by selenium supplementation and/or the selenium supplements restoring the activity of GPx or the level of antioxidants (GSH, ascorbate) [20,21,23,25,26].

The minor protection elicited by SeMet in the current in vitro study demonstrated an alternative mechanism of protection to those described previously. In the current study, the restoration of GPx activity that was lost on 40 µM HOCl insult was found to be a minor contributor to the protective effect of SeMet, with the main mechanism of protection elicited by SeMet owing to its ability to directly scavenge HOCl. This prevents the oxidant from reacting with cellular targets and mediating down-stream cellular dysfunction. It has previously been shown that SeMet can react with oxidants, including HOCl, at a comparable or greater rate than the rate of reaction of the oxidant with biological targets, therefore making SeMet a highly competitive substrate for the oxidant [17,50,51].

These data highlight a mechanism of protection that has significant implications in vivo, where the concentration of the therapeutic antioxidant, in this case SeMet, is required to be at a level that would allow it to compete for oxidants in an environment with a more abundant number of cellular targets. Hence, we also explored the effect of SeMet supplementation in vivo by using an experimental rat model of cardiac I/R injury. However, there was a lack of protection in overall cardiac function in both the acute and long-term in vivo model of cardiac I/R injury, and the accompanying adverse remodelling of the heart, despite the trend for the reduction in infarct size in SeMet-supplemented animals with I/R injury. The ability of selenium species to reduce infarct size has been shown previously in a similar in vivo model of cardiac I/R injury where the infarct was brought upon by the ligation of the left coronary artery [63] for 30 min and reperfusion for 60 min [30]. Thus, previously, Wistar rats fed a diet supplemented with sodium selenite (1.5 mg kg^−1^) for 10 weeks were shown to have a significantly smaller infarct size (33%), compared to animals fed a diet low in selenium (0.05 mg kg^−1^) [30]. In our current study, assessment of oxidative stress in vivo by measuring the level of plasma thiols also demonstrated no significant differences between control and SeMet-supplemented animals within the I/R injury cohort, which is in contrast to an experimental in vivo model of I/R injury that showed an improvement in the redox status of animals with I/R injury that were supplemented with sodium selenite (1.5 mg kg^−1^) in their feed for 10 weeks, as indicated by a significantly higher GSH:GSSG ratio in the plasma of these animals compared to animals fed a diet low in selenium (0.05 mg kg^−1^) [30].

The limited efficacy of SeMet observed in both our in vitro and in vivo studies contrast with previous studies that have demonstrated the benefits of using selenium supplementation in a therapeutic context. Much of the in vitro data relating to the efficacy of selenium species to prevent cellular damage, mainly through a reduction in the extent of cell death, support the potential beneficial effects of selenium supplements, yet these data were obtained in other cell types and/or models other than cardiac I/R injury, for example within the setting of diabetes [20,26,43,44]. Furthermore, animal studies that support the benefits of selenium supplementation in improving cardiac performance were performed in selenium-deficient animals and/or ex vivo models of I/R injury, i.e., Langendorff systems [45,46,47], and not within the setting of I/R as was the case here. Studies that employed Langendorff systems used hearts excised from animals fed either a normal diet or selenium-deficient diet (<0.02 mg kg^−1^ selenium), and achieved selenium supplementation either in the feed or by the infusion of sodium selenite (≥75 nM) solution prior to the induction of ischemia and/or during reperfusion [45,46,47]. Several indicators of cardiac performance, including left ventricular end diastolic pressure (LVEDP), rates of pressure development (+dP/dt) and pressure decay (−dP/dt) that diminished due to the induction of global ischemia and subsequent reperfusion were rescued in the hearts that were excised from animals that were supplemented with selenium.

The discrepancies between the current study and the literature may have several explanations. These discrepancies may be related to the fact that the animals used in the current study were selenium sufficient, suggesting that selenium supplementation may be more relevant in selenium-deficient individuals who are at high risk of experiencing a major adverse cardiac event (MACE). Langendorff systems used in previous studies may not also completely reflect the extent of oxidative stress found in vivo due to the absence of the inflammatory response, specifically the recruitment and migration of inflammatory cells into the infarcted myocardium, which can act as an additional source of oxidants that contribute to, and exacerbate, oxidative stress in the damaged heart tissue [3,5]. Hence, it is likely that the extent of SeMet supplementation in the in vivo model may not have been sufficient to completely combat the extent of oxidative stress generated. This concept was more evident in our in vitro study, where the concentrations of HOCl were in excess (two-fold) of the SeMet. This led to the inability of SeMet to compete with HOCl, and thus the resulting cellular dysfunction. Furthermore, studies have shown the non-specific incorporation of SeMet into proteins [48,49]. This may also account for the lack of efficacy observed in vivo, as this would limit the availability of SeMet either for selenoprotein synthesis and/or scavenging oxidant. Though our ICP-MS data confirmed significant elevations of tissue selenium in SeMet-supplemented animals, this did not reveal the type or level of selenium species present within the tissue.

## 5. Conclusions

Overall, we broadened the knowledge regarding the efficacy of selenium supplements in cardiac injury. Both the in vitro and in vivo data of our study suggest that there is an extent of damage which SeMet can protect against, and that the limited efficacy of SeMet may depend on the selenium status of individuals, as well as relate to the metabolism of SeMet and the biological activity of its metabolites. This has significant implications for the use of selenium supplements as a treatment for MI and/or HF and thus should be investigated further.

## Figures and Tables

**Figure 1 antioxidants-08-00546-f001:**
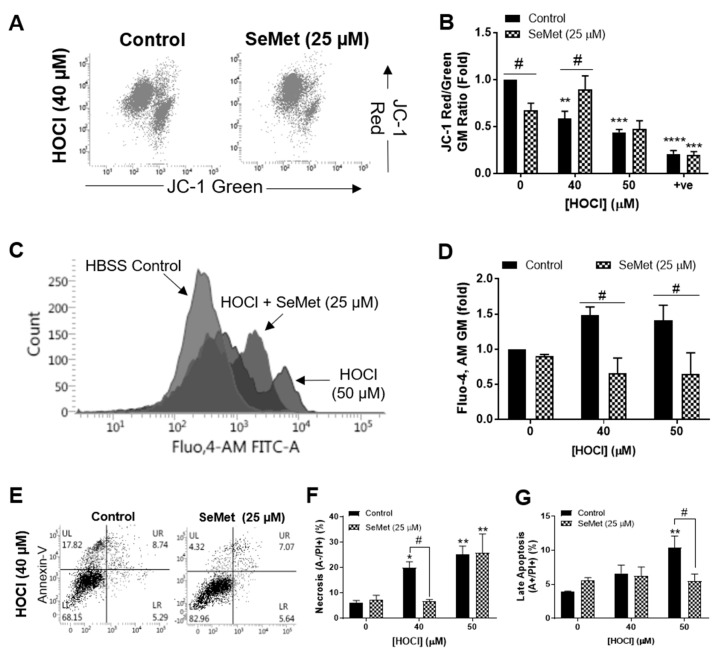
Selenomethionine (SeMet) supplementation protects against H9c2 cellular dysfunction elicited by HOCl. H9c2 cells (1 × 10^5^ cells mL^−1^) serum-starved and supplemented with (hatched bars) or without (solid bars) SeMet (25 µM) for 24 h were exposed to HOCl for 1 h with or without SeMet (25 µM) and analysed immediately for ΔΨ_m_ and intracellular Ca^2+^ accumulation using JC-1 and the fluorescent Ca^2+^ indicator Fluo,4-AM, respectively and the extent of cell death measured using Annexin-V/propidium iodide (PI) staining. (**A**) Representative dot plots of JC-1 staining with (**B**) quantification of ΔΨ_m_ immediately following exposure to HOCl and SeMet supplementation or CCCP (0.1 mM) as a positive control (+ve). (**C**) Representative histogram flow plot of Fluo,4-AM staining with (**D**) quantification of intracellular Ca^2+^ accumulation immediately following exposure to HOCl and SeMet supplementation. (**E**) Representative dot plots of Annexin-V/PI staining with (**F**) quantification of necrotic cells and (**G**) late apoptotic cells immediately following exposure to HOCl and SeMet supplementation. Data are expressed as a percentage of the whole cell populations as mean ± S.E.M. from *n* = 3 biological repeats performed in triplicate. (**B**,**D**) Data are expressed as a fold change over respective controls assayed in the absence of oxidant and SeMet as mean ± S.E.M. from *n* = 3 biological repeats performed in triplicate. * *p* < 0.05, ** *p* < 0.01, *** *p* < 0.001, **** *p* < 0.0001 vs. Hank’s buffered salt solution (HBSS) control (0 µM), # *p* < 0.05 vs. control cells assayed in the absence of SeMet (0 µM) as determined by two-way ANOVA with Bonferroni post-hoc testing.

**Figure 2 antioxidants-08-00546-f002:**
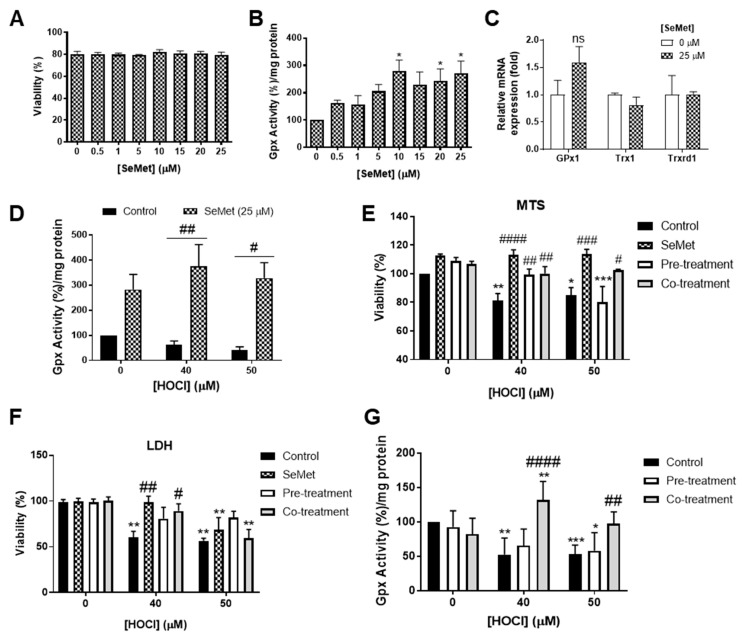
SeMet supplementation alters the redox status of H9c2 cells exposed to HOCl insult resulting in protection. (**A**–**C**) H9c2 cells (1 × 10^5^ cells mL^−1^) were exposed to either increasing concentration of SeMet in serum-free media for 24 h, (**D**) supplemented with SeMet (25 µM) prior to and during oxidative exposure and exposed to HOCl as described in Figure 1 or (**E**–**G**) exposed to HOCl alone (control) and subjected to different SeMet supplementation conditions (SeMet, supplementation throughout the whole treatment period; pre-treatment, supplementation prior to HOCl exposure; co-treatment, supplementation during HOCl exposure). (**B**,**D**,**G**) Quantification of glutathione peroxidase (GPx) activity immediately post treatment, with data expressed as a percentage fold change over control cells (0 µM) expressed as mean ± S.E.M (*n* ≥ 3). (**C**) mRNA gene expression of *GPx1*, *Trx1* and *Trxrd1* measured immediately post treatment using qPCR, with data expressed as a fold change of their respective controls as mean ± S.E.M (*n* = 3). Viability assessed (**A**) immediately or (**F**) 24 h post treatment using the lactate dehydrogenase (LDH) release assay or by using the (**E**) MTS assay immediately post treatment, with data expressed as a percentage viability relative to the control cells (0 µM) as mean ± S.E.M from *n* ≥ 3 biological repeats performed in triplicate. * *p* < 0.05, ** *p* < 0.01, *** *p* < 0.001 vs. HBSS controls; # *p* < 0.05, ## *p* < 0.01, ### *p* < 0.001, #### *p* < 0.0001 vs. control cells assayed in the absence of SeMet as determined by two-way ANOVA with Bonferroni post-hoc testing.

**Figure 3 antioxidants-08-00546-f003:**
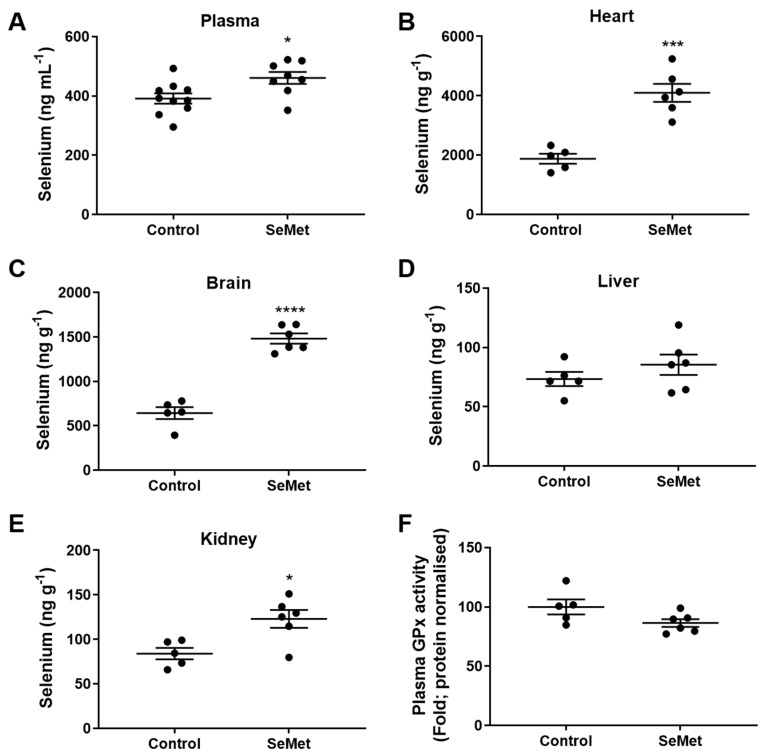
Selenium tissue levels in rats fed SeMet supplemented chow. Male Wistar rats (100–125 g) were randomly assigned into groups receiving either normal chow (control) or normal chow supplemented with SeMet (2 mg kg^−1^) ad libitum for 8 weeks. Quantification of selenium in the (**A**) plasma, (**B**) heart, (**C**) brain, (**D**) kidney and (**E**) liver was performed using inductively coupled plasma-mass spectrometry (ICP-MS) with (**F**) total GPx activity measured within rat plasma. Data expressed as mean ± S.E.M. from *n* = 5 (Control) and *n* = 6 (SeMet). * *p* < 0.05, *** *p* < 0.001, **** *p* < 0.0001 difference between control and SeMet-supplemented group as determined by two-tailed unpaired Student’s *t*-test.

**Figure 4 antioxidants-08-00546-f004:**
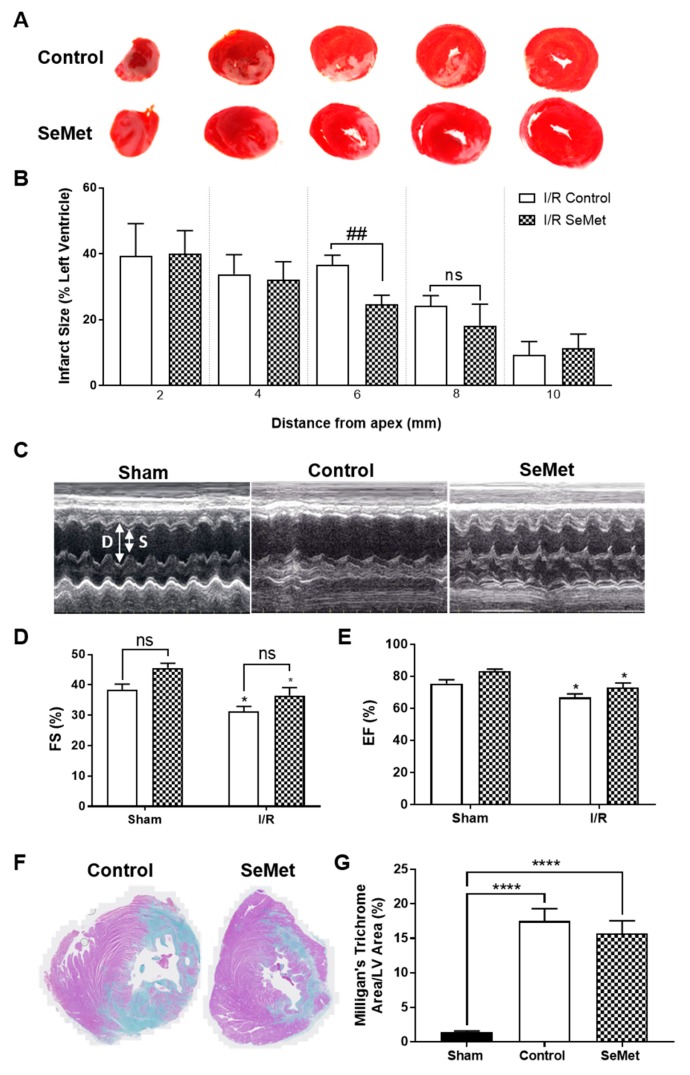
SeMet supplementation is not protective against cardiac ischemia/reperfusion (I/R) injury. Male Wistar rats (100–125 g) were fed either normal chow (control, open bars) or SeMet-supplemented chow (checked bars) as described in Figure 3 and subjected to 30 min ischemia or sham surgery followed by reperfusion and recovery for either 24 h or 4 weeks. (**A**) Representative images of 24 h heart left ventricle (LV) stained with triphenyl tetrazolium chloride (TTC) to differentiate infarcted tissue (white) from viable, muscle tissue (red) and (**B**) quantification of the infarcted region at 24 h expressed as the percentage area to the total area of the LV as mean ± S.E.M (*n* ≥ 5). (**C**) Representative echocardiogram images in M-mode and quantification (**D**) fractional shortening (FS) and (**E**) ejection fraction (EF) of control and SeMet-supplemented animals subjected to cardiac I/R injury. Data presented as mean ± S.E.M from control sham *n* = 5; control I/R *n* = 8; SeMet sham *n* = 5; SeMet I/R *n* = 8 rats per group, performed in triplicate. (**F**) Representative images of 4 week heart LV stained with Milligan’s trichrome to differentiate fibrotic tissue (blue) from muscle tissue (purple) and (**G**) quantification of the fibrotic region expressed as the percentage area to the total area of the LV as mean ± S.E.M sham *n* = 5; control *n* = 8; SeMet *n* = 8 rats per group, performed in triplicate. * *p* < 0.05, **** *p* < 0.0001 vs. sham, ## *p* < 0.01 vs. control, no significant (ns) changes between control and SeMet-supplemented group as determined by two-way ANOVA with Holm–Sidak post-hoc testing.

**Table 1 antioxidants-08-00546-t001:** Rat specific qPCR primers used in study.

Gene	Genebank Ref.	Forward Primer	Reverse Primer
Reference			
*Nono*	NM_001012356	CCTGATGCGAGAGAACAAGAGA	CTGGACGGTTGAATGCAGGA
*β-actin*	NM_012512.2	ATCAAGATCATTGCTCCTCCTG	CAGCTCAGTAACAGTCCGCC
Seleno-dependent antioxidants			
*GPx1*	NM_030826.3	CAGTCCACCGTGTATGCCTT	TGCCATTCTCCTGATGTCCG
*Trx1*	NM_053800.3	GCCCTTCTTTCATTCCCTCTGT	CTCCCCAACCTTTTGACCCTT
*Trxrd1*	NM_031614.2	CGTGCCGACGAAAATTGAAC	CATTGATCTTCACGCCCACG

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
