# Peer review of "Assessing the Efficacy of Dietary Selenomethionine Supplementation in the Setting of Cardiac Ischemia/Reperfusion Injury"

_antioxidants, 2019, doi:10.3390/antiox8110546_

Round 1

Reviewer 1 Report

The study by Leila Reyes et al. is an investigation on the effects of SeMet (selenomethionine ) against the oxidative cellular damage mediated by HOCl (hypochlorous acid ) in vitro in cardiomyocytes, as well as its potential beneficial effect in an experimental in vivo rat model of cardiac I/R (ischemia/reperfusion injury).

Through the analysis and integration of different parameters concerning the state of functional well-being of the cell, the authors arrive at a discrepant result with what is indicated by recent literature. Interestingly, both the in vitro and in vivo data of this study suggest that there is an extent of damage in which SeMet can protect against the cellular damage/death; however , the results here obtained in vivo indicated a limited efficacy of SeMet depending on the selenium status of individuals.

My criticism concerns the fact that observations with in situ morphological and histochemical techniques are very scarce. I realize that in this survey the chosen parameters are many and fundamental when dealing with studies on potential cytotoxicity or beneficial effect of substances. Then, the results obtained are more than sufficient for an unambiguous interpretation, although I hope the study will be expanded later as suggested.

These original data deserve to be published in the present form.

Author Response

The authors wish to thank the reviewer for their supportive feedback. We argree that there are indeed many other parameters that may be of interest, including in situ morphological/histochemical parameters, in studying the effects of selenomethionine supplementation. As the reviewer suggests, the current study is limited in this regard, however, further studies are ongoing.

Reviewer 2 Report

Manuscript ID: Antioxidants-640644

Title: Assessing the efficacy of dietary selenomethionine supplementation in the setting of cardiac ischemia/reperfusion injury.

This paper describes several tests performed on an in vitro cardiac myocyte model and an in vivo rat model of I/R heart injury. The authors intend to demonstrate whether there is any protective effect of a selenomethionine supplementation on this lesion.

Unlike other published studies, the authors have proved that this type of supplementation may have potential benefit in specific cases, but limited efficacy as a therapeutic agent for the treatment of heart attack.

The manuscript is concisely and clearly written, the presentation of data is good and well discussed. It should be emphasized that at the end of the discussion the authors present several hypotheses to justify the results obtained being different from those in the literature, which strengthens the discussion.

From the literature, it can be realized that the authors are experts in this subject, and this paper should represent an incentive for further research in this field.

Consequently, I recommend to accept this Article, after minor revision.

My observations are minor and given under:

Authors should mention the number of animals used in each group. In the graphs we notice a varying n, but we don't know how many animals were used effectively. The same is true for in vitro testing.

It would be convenient to reduce the captions of the figures.

Line 337 and 338_ specify the ”statistical significance”

Author Response

The authors thank the reviewer kindly for their comments and suggestions. All n values jave been amended in the Figure legends as follows:

All in vitro cell studies: data obtained from n 3 biological repeats performed in triplicate.

ICP-MS quantification of selenium levels in rat tissue is expressed as mean ± S.E.M. from n=5 (Control) and n=6 (SeMet).

Echocardiograpy: data is mean ± S.E.M from control sham n=5; control I/R n=8; SeMet sham n=5; SeMet I/R n=8 rats per group, performed in triplicate.

Histology data is expressed as mean ± S.E.M sham n=5; control n=8; SeMet n=8 rats per group, performed in triplicate.

We have kept the text of the figure legends as is, with the view that the detail assists the reader to follow the methods used to get the results presented. We are open to any editorial decision on the matter.

Line 337 and 338_ specify the ”statistical significance":  we have amended the text to read as follows: "...statistical significance (p < 0.05) demonstrated..."